# Boosting output performance of sliding mode triboelectric nanogenerator by charge space-accumulation effect

Wencong He[1,2], Wenlin Liu [1,2], Jie Chen[1], Zhao Wang[1], Yike Liu[1], Xianjie Pu[1], Hongmei Yang[1], Qian Tang[1], Huake Yang[1], Hengyu Guo [1✉] & Chenguo Hu [1✉]

The sliding mode triboelectric nanogenerator (S-TENG) is an effective technology for in-plane low-frequency mechanical energy harvesting. However, as surface modification of tribo-materials and charge excitation strategies are not well applicable for this mode, output performance promotion of S-TENG has no breakthrough recently. Herein, we propose a new strategy by designing shielding layer and alternative blank-tribo-area enabled charge space-accumulation (CSA) for enormously improving the charge density of S-TENG. It is found that the shielding layer prevents the air breakdown on the interface of tribo-layers effectively and the blank-tribo-area with charge dissipation on its surface of tribo-material promotes charge accumulation. The charge space-accumulation mechanism is analyzed theoretically and verified by experiments. The charge density of CSA-S-TENG achieves a 2.3 fold enhancement (1.63 mC m$^{-2}$) of normal S-TENG in ambient conditions. This work provides a deep understanding of the working mechanism of S-TENG and an effective strategy for promoting its output performance.

[1] Department of Applied Physics, State Key Laboratory of Power Transmission Equipment & System Security and New Technology, Chongqing University, 400044 Chongqing, P.R. China. [2] These authors contributed equally: Wencong He, Wenlin Liu. ✉email: cquphysicsghy@126.com; hucg@cqu.edu.cn

Harvesting energy from the ambient environment for self-powering distributed sensor networks has become a significant development direction in the Internet of Things (IoTs)[1–4]. Recently, based on the coupling of triboelectrification and electrostatic induction[5], the triboelectric nanogenerator (TENG) with the advantages of light weight, material variety, easy fabrication and low cost is attracting great attention[6–8] and has proved an efficient technology for harvesting low-frequency mechanical energy such as human motion[9–11], wind[12,13], water waves[14,15], etc.[16–18]. Generally, TENG can be divided into vertical contact-separation mode (CS-) and horizontal sliding mode (S-) based on the driving modes. Different from CS-TENG, S-TENG holds high efficiency[19,20], continuous and high output for in-plane regular movement (e.g. reciprocation and rotation) conversion[21–25], and it is a promising one towards commercialization[26,27]. Nevertheless, the low surface charge density is the bottleneck in the TENG output performance and its applications[28].

To boost the output performance, numerous research works have been carried out for enhancing the surface charge density of CS-TENG, such as the inner material optimization[29], surface physical/chemical modification[30], ion injection[31], environmental control[32], etc.[33–36]. A record high charge density of 2.38 mC m$^{-2}$ was achieved by charge excitation and quantifying contact in the latest work[37]. In these studies, researchers found that air breakdown effect occurred between the gap of tribo-surfaces during the separation process and the density of electron states on material surface are two key factors that limit the maximizing surface charge density of CS-TENG[32,38,39]. However, as surface modification of materials and charge excitations are not well applicable for S-TENG, the strategy to promote its output performance is rarely proposed and no breakthrough has been made since 2014. Zhu et al.[23] proposed radial arrays of micro-sized sectors on the contact surfaces, which enabled a high output power density of 19 mW cm$^{-2}$ at a rotation speed of 3000 rpm. Recently, Zhu et al.[40] reported a direct current TENG by using unidirectional transportation of charges and dual-intersection TENG, and successfully realized a continuous motion control in virtual space for next-generation real-time VR application in triboelectricity. Liu et al. reported a constant current S-TENG and indicated that air breakdown effect would happen on the sliding edge of two tribo-layers. By directly utilizing the discharged charges and improving contact, this kind of device reached 460 μC m$^{-2}$ charge density compared with 70 μC m$^{-2}$ from controlled S-TENG device[41]. In most cases of S-TENG, air breakdown happens not only on the edge of sliding layer, but also in the overlapped interface due to inescapable air voids between two osculatory trio-layers[42]. Therefore, taking air breakdown into consideration, developing strategies for maximizing the surface charge density for S-TENG is highly desired.

Herein, we propose a new design of S-TENG, the charge space-accumulation S-TENG (CSA-S-TENG) for improving surface charge density. CSA-S-TENG incorporates a shielding layer on a slider and an alternating blank-tribo-area structure with charge dissipation on a stator. Based on a grounded conductive layer cover on the back of the slider, air breakdown can be contained to a great extent, and by further introducing an extra blank-area structure with charge dissipated tribo-material on the stator, charge space-accumulation effect is achieved. The theoretical mechanism of the charge transfer in each CSA-S-TENG cycle is proposed and analyzed, and the mechanism is also verified by experimental data. The factors that influence the effective charge density of CSA-S-TENG are systematically studied as well. With optimized structure and materials, the charge density of CSA-S-TENG achieves a 2.3-fold enhancement (1.63 mC m$^{-2}$) of normal S-TENG in ambient conditions. Meanwhile, the shielding

electrode also contributes to the whole charge output as an independent power outlet. For demonstration, a rotary CSA-S-TENG is fabricated with a diameter of 10 cm and the maximum output power density reaches 1.15 W m$^{-2}$ Hz$^{-1}$, three times as that of previous reports[23]. The rotary CSA-S-TENG is also used to charge a 10 μF capacitor to 5 V in 15 s and directly power over 912 LEDs at 1 Hz. This work provides a facile and effective strategy of boosting charge output of S-TENG and also presents core factors for the high-performance S-TENG design.

## Results

**Charge space-accumulation effect and working mechanism.** In the traditional S-TENG, air voids are inevitably formed between two tribo-layers where air breakdown occurs, which largely limits the surface charge density and output performance. In this work, considering the charge limitation of air breakdown and device structure and materials design, a newly designed sliding mode TENG is proposed by creating a grounded shielding electrode on the back of the slider and introducing an extra blank-tribo-area on the stationary part to achieve charge accumulation called CSA-S-TENG. This new S-TENG is quite different from the traditional S-TENG in structure. Figure 1a shows the 3D structural scheme of the basic device. Here, polytetrafluoroethylene (PTFE) and nylon (PA) film were used as tribo-materials. To further reduce the formation of air voids, sponge foam was used to optimize the contact status between tribo-layers. Insets 1 and 2 depict the cross-section schematic and top-view photograph of the stationary part and sliding part, respectively. The detailed fabrication process of the device is presented in the "Methods" section. For conventional S-TENG, due to the limitation of air breakdown, 2Q transferred charge quantity generates in half a sliding period as the working mechanism and charge distribution schematically premised in Fig. 1b. The transferred charge quantity ($Q_t$) is equal to the difference value of two bottom electrodes as shown in Eq. (1).

$$Q_t = Q_{LBE} - Q_{RBE}, \quad (1)$$

where $Q_{LBE}$ is the charge quantity on left bottom electrode (LBE) and $Q_{RBE}$ is the charge quantity on right bottom electrode (RBE).

When the grounded electrode on sliding layer is introduced, the potential distribution in the air voids is significantly changed. As shown in Fig. 1c, the simulated results indicate three times the potential difference between top (PTFE) layer and bottom (PA) layer with and without shielding layer, which could greatly avoid air breakdown and hold larger charge density on the tribo-layers. The grounded electrode can indeed increase the surface charge density of the tribo-layers. However, the presence of shielding layer also weakens the electrostatic induction effect on bottom electrodes, thus affecting the output charge quantity. As can be seen from the first two cases (the shielding electrode of S-TENG is grounded/ungrounded) in Fig. 1j, the effective charge output between two bottom electrodes of S-TENG is rarely improved whether the shielding electrode is grounded or not, which also agrees with a previous report[43]. According to the formula of plate capacitance, charge conservation and the capacitance model of S-TENG device, we can get the relationship of transferred charge $Q_t$ and charge distribution on tribo-layers (Eq. (2)).

$$Q_t = \frac{Q_0 - Q_R}{\frac{d_2 \varepsilon_{r1}}{d_1 \varepsilon_{r2}} + 1} - Q_R, \quad (2)$$

where $Q_0$, $d_1$ and $\varepsilon_{r1}$ are charge quantity, thickness and relative permittivity of PTFE, respectively; $d_2$ and $\varepsilon_{r2}$ are thickness and relative permittivity of PA, respectively; $Q_R$ is the corresponding charge of PA tribo-layer on RBE. The detailed calculations and analysis on the charge distribution of the shielding layer,

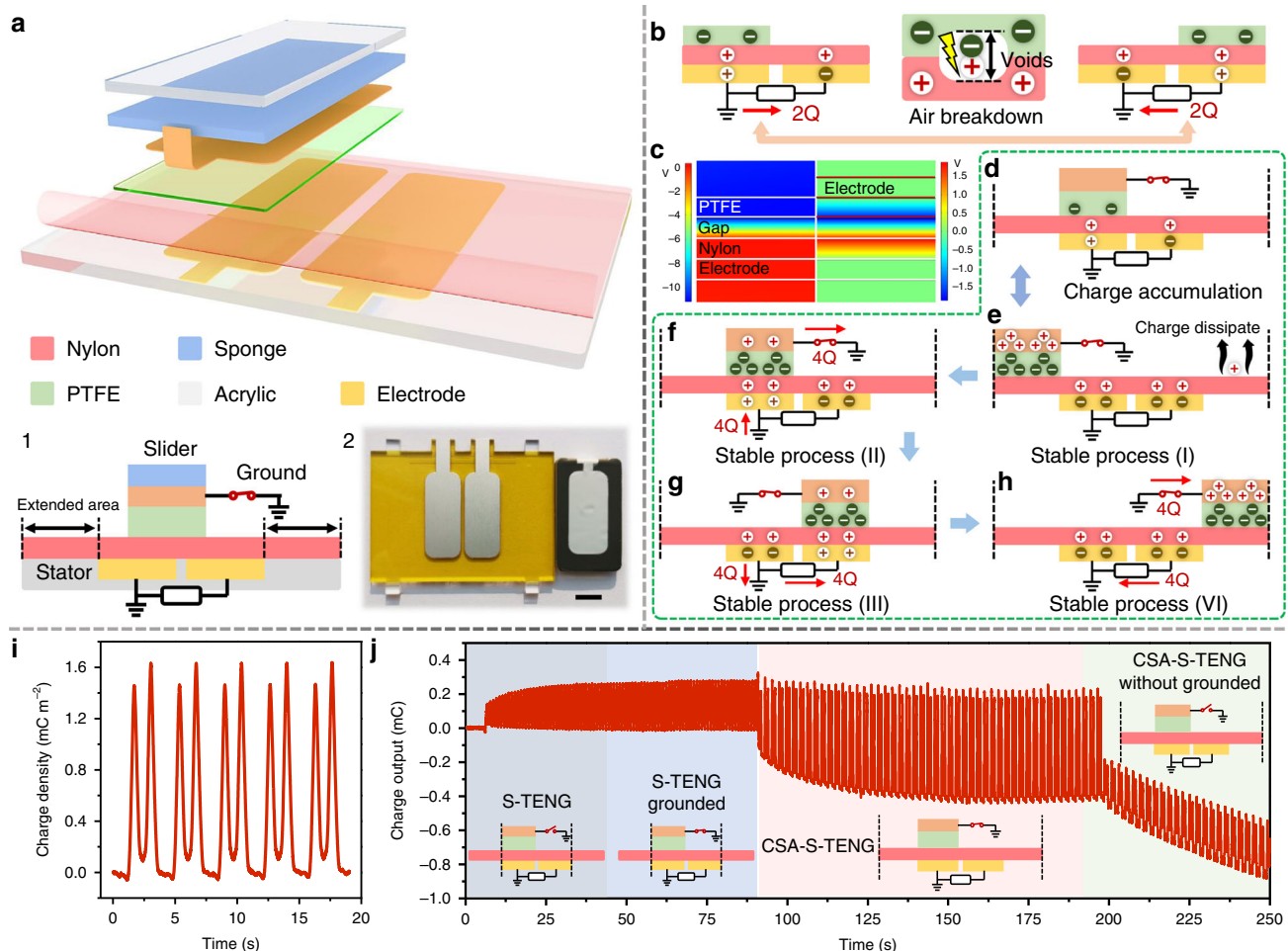

**Fig. 1 Structure and mechanism of CSA-S-TENG. a** The 3D schematic of CSA-S-TENG. Inset 1. Cross-section view shows the extended area structure and grounding electrode. Inset 2. Photograph of the device. The area of top electrode is 5 cm². The effective area of the stationary part is 26 cm². Scale bar: 1 cm. **b** Working mechanism and air breakdown of traditional S-TENG. **c** Potential simulation of air gap existence in CSA-S-TENG with/without shielding electrode. **d** Schematic of CSA-S-TENG with the same charge distribution as traditional S-TENG in the initial stage. **e** Stable charge distribution of CSA-S-TENG after multiple sliding cycles; charge generated on extended area dissipates into air, due to that there is no back electrode below to bound charge. **e–h** Working mechanism and relative charge transfer quantity of CSA-S-TENG in four states during half working period. **i** The output charge density curve of CSA-S-TENG. **j** Dynamic output charge curve under four working circumstances.

electrodes and tribo-layers, and the charge transfer between the bottom electrodes with or without a shielding layer are given in Supplementary Fig. 1 and Supplementary Note 1.

Based on the experiment results and theoretical analysis, the relatively low saturated surface charge density on PA film in this structure is the key factor that prevents the charge of both tribo-layers from further accumulating. Therefore, we propose an extra blank-tribo-area besides two electrodes on the stator to extend the motion range of the slider, to achieve high surface charge density by the charge space-accumulation effect. Figure 1d shows the schematic charge distribution of CSA-S-TENG at the initial stage evolving from S-TENG. Benefiting from shielding electrode and extra blank-tribo-area structure, after several sliding cycles, the surface of top tribo-layer (PTFE) accumulates charges from tribo-surface on the bottom electrode and extra tribo-area respectively, and form a stable charge distribution state as shown in Fig. 1e.

It is worth noting that, without the bottom electrode for equilibrating electrostatic field and the feature of PA material, the most charge on the blank-tribo-area would be quickly dissipated, which ensures continuing charge replenishment on PTFE during each friction (Supplementary Figs. 2, 3), so the CSA-S-TENG can easily achieve stable and multifold output charge. The process of

this charge space-accumulation is analyzed in detail and presented in Supplementary Fig. 4 and Supplementary Note 2, and the charge accumulation process of CSA-S-TENG will go on for several cycles. Figure 1e–h shows the working mechanism and four charge distribution states in half a cycle of CSA-S-TENG when surface charge density reaches stable, which indicates that 4Q transfer charge quantity can be generated from the bottom electrodes as well as the shielding electrode. The output of shielding electrode will be separately discussed in the following part. The detailed working mechanism and charge transfer process are illustrated in Supplementary Fig. 5 and Supplementary Note 3. The third case (red background area) in Fig. 1j shows the dynamic charge output curve of CSA-S-TENG with the grounded shielding electrode (more details shown in Supplementary Fig. 6 and Supplementary Movie 1). The charge space-accumulation process can be observed and the charge output quickly reaches stable state, which is consistent with our theoretical analysis. With experimental optimization, the stable output charge density reached 1.63 mC m⁻², and the basic charge output curve is shown in Fig. 1i. Additionally, in Fig. 1j (green background area), when we cut off the grounding state of CSA-S-TENG, a rapid decline in the charge output is observed, which

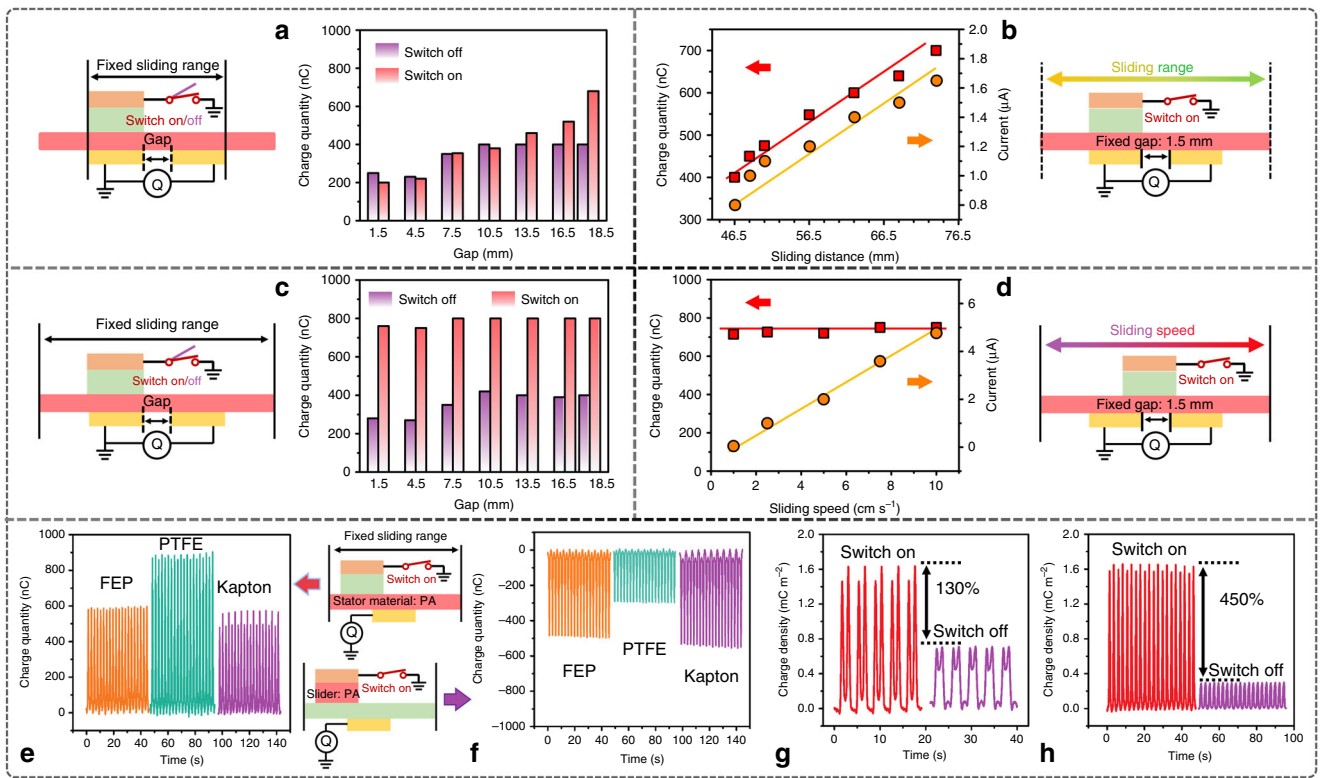

**Fig. 2 Structure and materials' influence on the output of CSA-S-TENG. a** Output charge of double-bottom-electrode S-TENG with different inner electrode gaps when grounded switch is OFF/ON. **b** Output charge and current of CSA-S-TENG with fixed electrode gap while varying sliding range (speed fixed at 4 cm s$^{-1}$). **c** Transferred charge of CSA-S-TENG with different inner electrode gaps when grounded switch is OFF/ON. **d** Output charge and current of CSA-S-TENG with fixed bottom electrode gap while varying sliding speed (range fixed at 76.5 mm). **e**, **f** Output charge of single electrode CSA-S-TENG with fixed stationary material PA while varying sliding materials (FEP, PTFE and Kapton) and fixed sliding material PA while varying stationary materials (FEP, PTFE and Kapton), respectively. Optimized output charge density of **g** double electrode and **h** single electrode CSA-S-TENG with grounded switch OFF/ON.

can also prove the inhibitory effect of the shielding electrode on air breakdown. Based on the proposed principles above, the CSA-S-TENG in two modes is designed: the plate sliding mode CSA-S-TENG and rotary CSA-S-TENG. Supplementary Figure 7 shows the photographs of those devices.

**Influence of the structure and material**. According to above theoretical analysis and preliminary experimental results, a series of experiments were carried out for further investigations of the structure's and material's influence on the charge space-accumulation effect of CSA-S-TENG.

Firstly, the output charge of traditional double-bottom-electrode S-TENG with different electrode gap was measured when the grounded switch is in on and off status respectively, as schematically shown in Fig. 2a. The increase in gap distance provides larger blank-tribo-area for charge space-accumulation process. In this case, with the use of the shielding electrode, a large amount of charge would accumulate on the PTFE surface, while the saturated maximum surface charge would be highly limited by the inner air-break-down effect without the shielding electrode. From the test results in Fig. 2a, with the increase of gap distance from 1.5 to 18.5 mm, the output charge increases linearly from 200 to 700 nC for S-TENG with the shielding electrode and quickly becomes saturated from 200 to 400 nC for S-TENG without the shielding electrode. Furthermore, as shown in Fig. 2b, with fixed gap distance, the increase of blank-tribo-area and sliding range on the outside area of the two bottom electrodes also leads to the enhancement of charge space-accumulation effect in CSA-S-TENG. With the length of extra tribo-area

extending from 46.5 to 76.5 mm, the output charge and current of CSA-S-TENG increase from 400 nC, 0.8 μA to 700 nC, 1.6 μA linearly, and the optimized ratio for maximum performance is different, 26.23 and 20.68 nC mm$^{-1}$ for gap and extra sliding range, respectively, due to the different influence caused by the electric field of bottom electrodes.

And when the blank-tribo-area is fixed the same large as the slider area, the addition of gap distance rarely affects the output charge which is saturated at 800 nC as plotted in Fig. 2c and the current of CSA-S-TENG with different gaps at fixed sliding rang as shown in Supplementary Fig. 8. Above results and discussion reveal that the shielding electrode and an optimized blank-tribo-area are both necessary for maximizing the charge space-accumulation effect. In addition, with fixed blank-tribo-area and gap distance, the effect of sliding speed on charge output of CSA-S-TENG is measured. As depicted in Fig. 2d, the sliding speed shows no obvious relationship with the final saturated charge output and the output keeps at ~800 nC from 1 to 10 cm s$^{-1}$. Therefore, CSA-S-TENG is also appropriate for low-frequency mechanical energy harvesting, which can ensure the device mechanical durability while maintaining high output charge density at the same time.

Secondly, for simplification, we systematically measured the output of the single-bottom-electrode CSA-S-TENG with varying the slider and stator tribo-materials. Similarly, sliding distance and velocity have the similar effect on single-bottom-electrode CSA-S-TENG (Supplementary Fig. 9). In Fig. 2e, we used nylon (PA) as the stator material, and tested the charge output of CSA-S-TENG while using different slider material like fluorinated

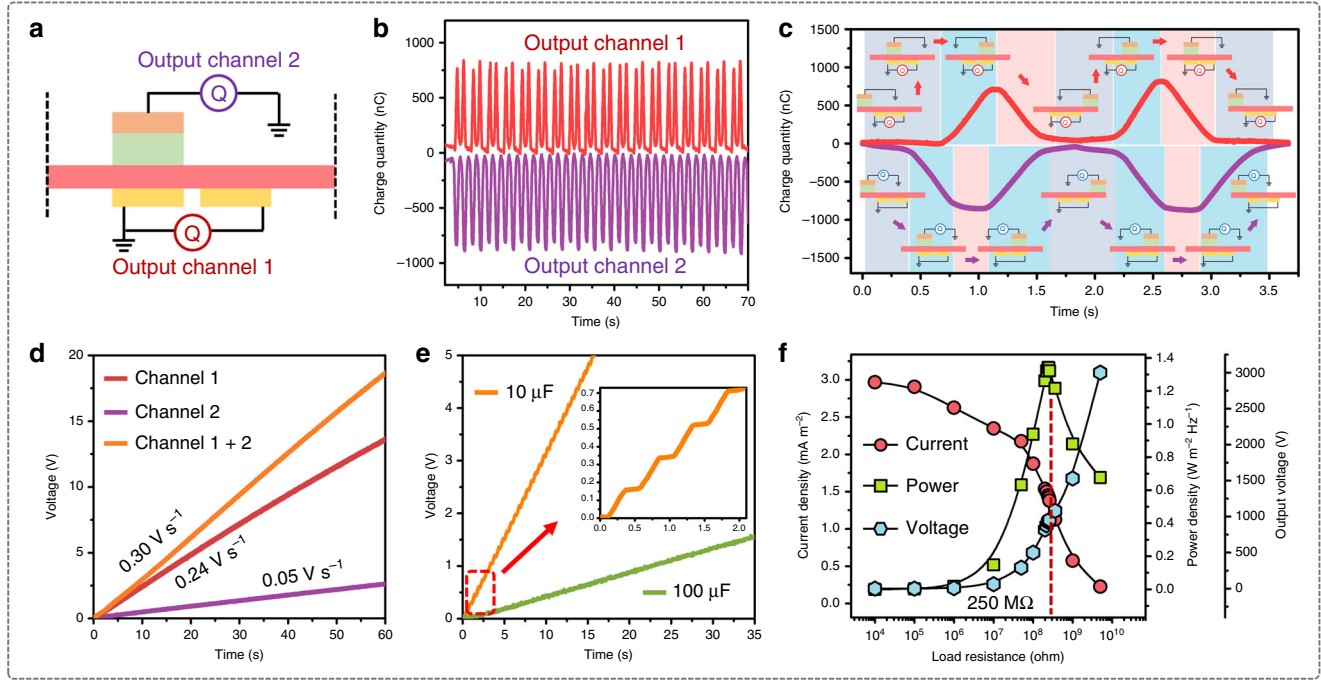

**Fig. 3 Performance of double output channel in CSA-S-TENG. a** Test schematic of the double output channel. **b** Simultaneously recorded output charge from channel 1 (red) and channel 2 (purple). **c** Enlarged output charge curves and corresponding working process of two channels during one sliding cycle. **d** Voltage curves of charging 10 μF capacitor at 1 Hz using channel 1, channel 2 and both. **e** Use channel 1 for charging 10 and 100 μF capacitor at 1 Hz individually; inset shows the enlarged step-like charging character. **f** Matching impendence measurement of CSA-S-TENG.

ethylene propylene (FEP), polytetrafluoroethylene (PTFE) and polyimide (Kapton). The results show that PTFE and PA pair yield the highest output, which indicates that PTFE holds the highest surface charge state density than the other materials. Correspondingly, when using PA as slider tribo-material, we measured the charge output of CSA-S-TENG with FEP, PTFE and Kapton as the stator material, respectively. Differently, in Fig. 2f, PA and PTFE pair presents the lowest output. It indicates that PTFE holds the strongest property of keeping charges and makes the charge on blank-tribo-area dissipate slowly, which impedes the charge space-accumulation effect of PA layer (as measured in Supplementary Figs. 3, 10, PA material also exhibits the fastest charge dissipated character). Furthermore, the different thickness of PTFE within a certain range has little effect on the output due to the replenishment ability of extra blank-tribo-area (as shown in Supplementary Fig. 11). Therefore, thicker dielectric materials can be used as the triboelectric layer for durability. Thus, for the slider material, we would better choose charge-keeping property and higher surface charge state density, and for the stator material, we need a faster charge dissipating feature to maximize charge space-accumulation effect. The output of both single- and double-bottom-electrode CSA-S-TENG can be optimized (Supplementary Figs. 12, 13).

From the above analysis and results, it is clear that the existence of the shielding electrode, the size of blank-tribo-area and materials all contribute to the charge space-accumulation effect. Finally, with optimized structure and materials, the output charge density of the double-bottom-electrode and single-bottom-electrode CSA-S-TENG both achieve 1.63 mC m$^{-2}$, which is 2.3 and 5.5 times of the traditional devices (as shown in Fig. 2g, h).

**Double-output-channel feature.** Based on the working mechanism in Fig. 1e–h and Supplementary Fig. 5, there would be the same amount of opposite sign charge transfer as bottom electrodes between the shielding electrode and the ground during operation period, which indicates that the shielding electrode could contribute to the whole electric output performance of CSA-S-TENG as an independent channel. Here, as schematically shown in Fig. 3a, we set the bottom electrodes and shielding electrode as output channel 1 and output channel 2 respectively, and measured their output charge during sliding cycles simultaneously. The recorded output charge curves are presented in Fig. 3b. Experimental results verify that nearly same amount of opposite sign (phase) charges are generated from channel 1 (830 nC) and channel 2 (850 nC), which is highly consistent with the theoretical analysis. The detailed dynamic charge output curves of two channels with corresponding sliding position in one working cycle are synchronously illustrated in Fig. 3c, which also well match with our proposed mechanism. And the double output has little distinction with the different gap, in particular, the output of channel 2 (Supplementary Fig. 14).

To demonstrate the output performance of double-output-channel CSA-S-TENG, they were used separately and in combination to charge a 10 μF commercial capacitor after rectifying (the connection of each channel in rectifier circuit is illustrated in Supplementary Fig. 15). Figure 3d presents the voltage charging curves of these channels, and the charging rate of combined channels (0.3 V s$^{-1}$) satisfies the summation of channel 1 (0.24 V s$^{-1}$) and channel 2 (0.05 V s$^{-1}$), which exhibits the contribution of the shielding electrode as the electric outlet. Due to the synchronous output (in opposite phase) of the two channels, the charging curve is uniformly stepwise, as shown in the inset of Fig. 3e. (The big voltage steps of 1 μF capacitor (at 69% humidity) indicate the high charge density as shown in Supplementary Fig. 16 and Supplementary Movie 2.) Furthermore, an excellent humidity adaptability is also verified for CSA-S-TENG by measuring output charge in different humidity from 10 to 80% RH (Supplementary Fig. 17). It is worth noting that, since channel 2 is equivalent to a single electrode mode TENG,

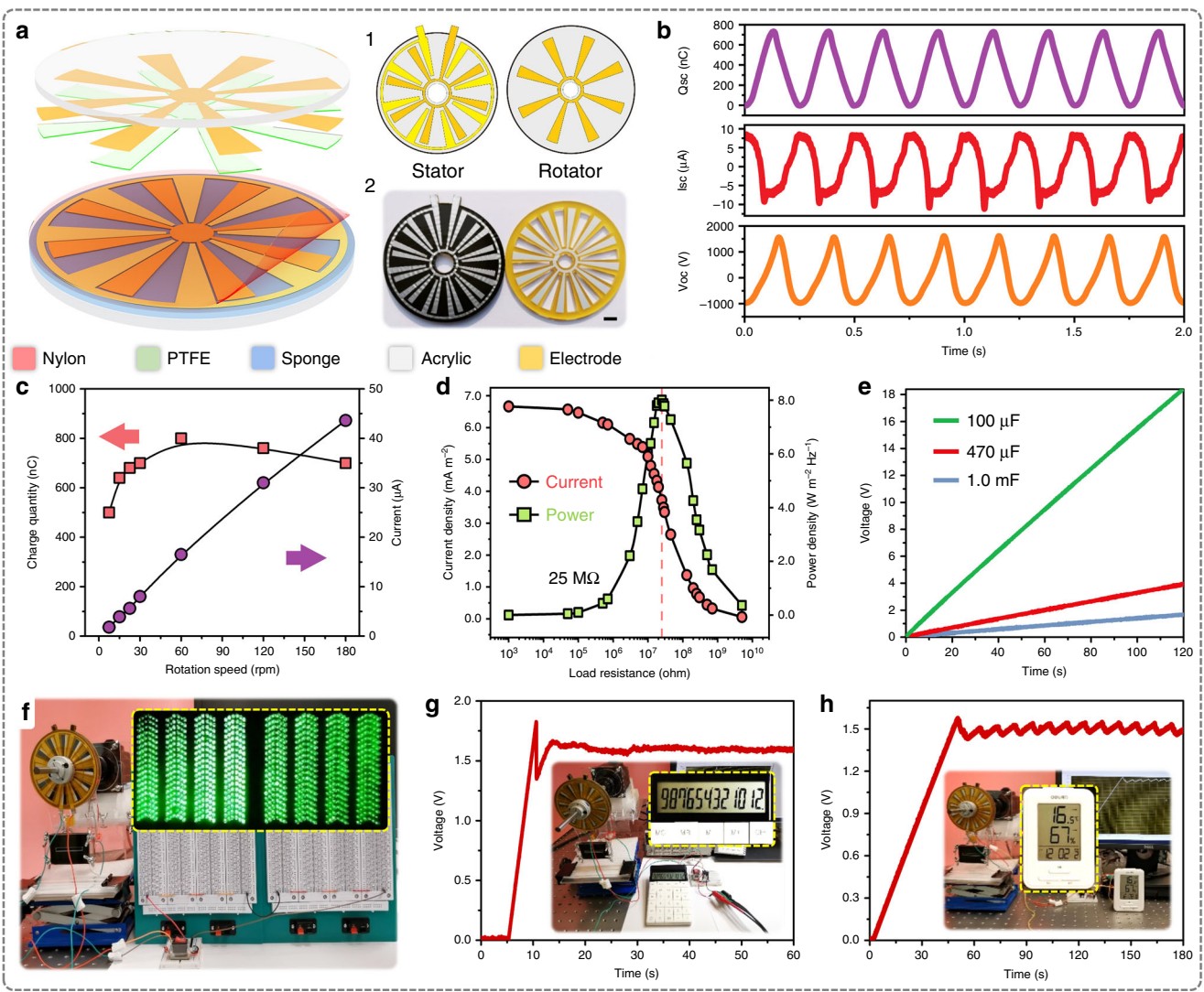

**Fig. 4 Performance and application of rotation-type CSA-S-TENG. a** 3D structural schematic of the rotary device. Insets 1 and 2 respectively depict the top-view schematics and device photographs of stator and rotator part. Scale bar: 1 cm. **b** Short-circuit charge ($Q_{sc}$), short-circuit current ($I_{sc}$) and open-circuit voltage ($V_{oc}$) of rotary CSA-S-TENG at 0.5 Hz working speed. **c** Transferred charge and current of rotary CSA-S-TENG at different rotational speed. **d** Matching impendence and output power evaluation of rotary CSA-S-TENG at 1 Hz working speed. **e** Voltage curves of charging 100 μF, 470 μF and 1 mF capacitor using rotary CSA-S-TENG at 1 Hz speed. **f** Directly driving 912 LEDs at 60 rpm. **g** Charging 22 μF capacitor while powering a scientific calculator at 110 rpm. **h** Charging 470 μF capacitor while powering a hydro-thermometer at 120 rpm.

although the short-circuit charge value of channel 2 is the same as channel 1, the output capability largely shrinks. Moreover, in Fig. 3e under 1 Hz sliding frequency, the double-channel CSA-S-TENG can charge 100 μF capacitor to 1 V in only 20 s with channel 1. Finally, the electric measurement of double-channel CSA-S-TENG under various external load resistance was performed for evaluating its matching impendence and maximum output power. As shown in Fig. 3f, the maximized output power density reaches 1.3 W m$^{-2}$ Hz$^{-1}$ at 250 MΩ external load, which is rather high for a single S-TENG unit.

**Demonstrations in rotary form**. The most significant application of the sliding mode TENG is to harvest regularly rotational movement. Based on a basic sliding TENG unit, CSA-S-TENG can be easily designed to rotational working mode as 3D schematically depicted in Fig. 4a. Different from the conventional rotational S-TENG device (Supplementary Fig. 18a), there exists alternative blank-tribo-area among radially interdigital electrode

and the shielding electrode behind sliding layer for rotary (R-) CSA-S-TENG. Insets 1 and 2 show the top schematic view and photograph of the stator and rotator, respectively, where the diameter of the disk and effective contact area are 10 cm and 11.27 cm$^2$ respectively. The short-circuit charge, current and open-circuit voltage curves of R-CSA-S-TENG under 30 rpm rotation speed are presented in Fig. 4b. As a comparison, under the same pressure force between the stator and rotator, the output charge density of R-CSA-S-TENG (730 μC) is much higher (3.17 times) than the conventional R-S-TENG (230 μC) with almost the same output curve form (Supplementary Fig. 18b).

In Fig. 4c, under different rotation speed, R-CSA-S-TENG delivers 500 nC, 1.8 μA at 7.5 rpm and 700 nC, 43.6 μA at 180 rpm, which ensured its capability of harvesting mechanical rotation energy in ultra-low and moderate frequency range. In addition, Fig. 4d shows the output power and current density at different resistance from 1 kΩ to 5 GΩ, and the maximum power density reaches 8 W m$^{-2}$ Hz$^{-1}$ (divided by effective contact area) and 1.15 W m$^{-2}$ Hz$^{-1}$ (divided by whole disk area) with

matching impedance of 25 MΩ, which is three times as that of previous report of the rotary TENG with fine micro-sized sectors on the contact surfaces[23]. To demonstrate its potential application, R-CSA-S-TENG is first used to charge capacitors. The voltage curves at the rotation speed of 60 rpm are shown in Fig. 4e, where the 100 μF, 470 μF and 1 mF capacitors can be charged to 18, 4 and 1.7 V within 120 s, respectively.

Furthermore, the R-CSA-S-TENG can power some electrical devices at low speed by using different capacitor and full-wave rectifier. A total of 912 green light-emitting diodes (LEDs) with a diameter of 5 mm in series can be lit up directly at the low speed of 60 rpm (Fig. 4f and Supplementary Movie 3). Figure 4g shows the R-CSA-S-TENG sustainably power the digital scientific calculator with 22 μF capacitor at a speed of 110 rpm. The voltage of the capacitor connected with calculator is monitored by a voltmeter. Then, the capacitor is changed into 470 μF for more complicated calculation as shown in Supplementary Movie 4. Lastly, a commercial multifunctional thermo-hygrometer can be powered by R-CSA-S-TENG with 470 μF capacitor at the speed of 120 rpm (Fig. 4h and Supplementary Movie 5). As the thermo-hygrometer collects the data of external temperature and humidity in every 10 s, some fluctuations in the voltage curve indicate that it is regularly working.

## Discussion

In summary, we analyzed the essential mechanism of surface charge density, and found that air breakdown effect and material surface electron states are the key factors for the output limitation for the sliding mode TENG. Consequently, we proposed a strategy of using the shielding electrode for containing air breakdown in inner voids, and realizing the charge space-accumulation effect by further adding the alternative blank-tribo-area, which largely boosted the surface charge density and output power density of CSA-S-TENG. The charge dissipation of materials was also proved to affect the charge accumulation. A highest 1.63 mC m$^{-2}$ output charge density was achieved through the structure and material optimization, which was 2.3 times that of a conventional S-TENG. Moreover, we also evaluated the output contribution of the shielding electrode as an independent channel to the whole CSA-S-TENG device. Lastly, rotary CSA-S-TENG was fabricated and its great potential in low-frequency rotation energy harvesting and self-powered systems was demonstrated. This work provided insights and a promising avenue for boosting charge output density of the sliding mode TENG.

## Methods

**Fabrication of the double-bottom-electrode CSA-S-TENG**. Stator: (i) A rectangle acrylic sheet with a dimension of 7.75 cm (length) by 5.4 cm (width) by 3 mm as the substrate was cut by a laser cutter. (ii) Two chamfered 4 mm, 15 mm by 35 mm by 15 μm Al electrodes were adhered to the middle of the surface on the substrate with a gap between them. Both Al electrodes are connected by conductive wires for electrical measurement. (iii) A 25-μm thickness PA film of the same dimensions with the substrate was adhered to the upper surface of the Al electrodes and substrate as the positive triboelectric layer. Slider: (i) A rectangle acrylic sheet with a dimension of 2.8 cm by 4.6 cm by 3 mm as the substrate was cut by a laser cutter. (ii) A 20 Psi foam with the dimension of 2.8 cm by 4.6 cm by 1 mm was adhered to the surface for optimizing the contact. (iii) A chamfered 4 mm, 15 mm by 35 mm by 15 μm Al electrode was adhered to the upper surface of the foam. (iv) A PTFE film with a dimension of 2.8 cm by 4.6 cm by 50 μm was adhered to overall surface of Al electrode and the foam. The Al electrode is grounded through a wire with a switch.

**Fabrication of the single-bottom-electrode CSA-S-TENG**. Stator: (i) A rectangle acrylic sheet with a dimension of 7.75 cm (length) by 5.4 cm (width) by 3 mm as the substrate was cut by a laser cutter. (ii) One chamfered 4 mm, 15 mm by 35 mm by 15 μm Al electrode was adhered to the middle of the surface on the substrate. Al electrode is connected by conductive wires for electrical measurement. (iii) A 25-μm thickness PA film of the same dimensions with the substrate was adhered to the upper surface of the Al electrode and substrate as the positive triboelectric layer.

Slider: The fabrication is exactly same as the slider of the double-bottom-electrode CSA-S-TENG.

**Fabrication of the rotatory CSA-S-TENG**. Stator: (i) A disk-shaped acrylic as the substrate was cut using a laser cutter, which has a diameter of 10 cm and a thickness of 3 mm. (ii) A hole with a diameter of 1.6 cm at the center of the substrate was drilled for placing a bearing in. (iii) Shallow ditches on the surface of the substrate were made by a laser cutter. These ditches define the patterns of two sets of complementary radial-arrayed Al electrodes (length: 2.8 cm) and free frictional area. The Al electrode, the inset gap and EA with the central angle of 6.5°, 4°, and 13°, respectively. (iv) An Al electrode was adhered to the surface of the substrate and cut along the ditches. All Al electrodes were connected by conductive wires for electrical measurement. (v) A 20 Psi foam was adhered between the Al electrode and the substrate with a diameter of 10 cm and a thickness of 1 mm. (vi) A PA film with the thickness of 25 μm was adhered to the whole surface of Al electrodes and foam as the triboelectric layer. Rotator: (i) A disk-shaped acrylic was cut as the substrate with a diameter of 10 cm and a thickness of 3 mm. (ii) A hole with a diameter of 1.2 cm was made at the center of substrate, and sectors were cut and removed with a central angle of 11.5° and a length of 3.2 cm around center on the disk. (iii) The patterns of the radial-arrayed electrode with the same length were cut using laser cutter. (iv) An Al electrode was adhered to the surface of the substrate and cut along the patterns. The Al electrode with the central angle of 8.5°, making sure the effective contact area. (v) A PTFE film with a diameter of 10 cm and a thickness of 50 μm was adhered to the surface of electrodes and the substrate.

**Electrical measurement and characterization**. All devices were fixed on an optical table (ZPT-G-M-15-10) for measurement. The sliders of CSA-TENG and the normal CS-TENG were driven by a linear motor (LinMot PS01-37×120-C). The rotary process was performed by a commercial programmable stepper motor (86BYG250D). The short-circuit current, the transferred charges and the voltage of capacitors were measured by an electrometer (Keithley 6514), and the load voltage by a high-speed electrostatic voltmeter (Trek model 370) with series resistance voltage division method. The room temperature (10−25 °C) and humidity (58o−85% RH) were measured by commercial thermo-hygrometer (TH20R).

## Data availability

The data that support the plots within this paper and other findings of this study are available from the corresponding authors upon reasonable request.

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

## Acknowledgements

This work was supported by the National Natural Science Foundation of China (NSFC) (51572040, 51902035, 51772036) and the Fundamental Research Funds for the Central Universities (2019CDXZWL001, 2020CDCGJ005).

## Author contributions

W.H., W.L. and C.H. conceived the idea, designed the experiments and analyzed the data. W.H. fabricated the devices and performed the electrical performance measurement. W.H. and W.L. performed Supplementary Materials. Y.L., X.P., Z.W., Hongmei Y., Q.T., Huake Y. and J.C. provided some suggestions on experiment. W.H., C.H., H.G. and W.L. wrote the manuscript. H.G. and C.H. supervised the project. All authors discussed the results and contributed to the manuscript.

## Competing interests

The authors declare no competing interests.
