## [Peer Review File · Nature Communications]

Reviewers' Comments:

Reviewer #1:

Remarks to the Author:

1. What does 'in-plane low frequency mechanical energy harvesting' mean? Why is it the 'most effective' method? What are the comparing metrics here? Please provide more detailed discussions.
2. The authors emphasized several times that TENG is the 'most effective', 'most promising' energy harvester in different applications. However, there is no data or chart to support this argument directly. Please benchmark using relevant figures of merits to help readers understand it.
3. How good is '2.3 times'? How to evaluate this value? It is easy to boost the charge density from a tiny number to a relatively large number; however, it is challenging to improve the charge density starting from a big number.
4. Is this 1.63 mC/m^2 physically correct? Please justify this number by adding some physical explanation. What is the mechanism of keeping these charges on the surface, and most importantly, how to effectively measure the value?
5. What is the scientific meaning of the FEA result in Fig. 1c? What is the boundary condition in the FEA initial setup? What is the contribution of the FEA result in this paper?
6. In Fig. 1b and also supporting Fig 1, the charge distribution in Nylon and metal is the same (both positive). Is it reasonable? If so, it means there is no electrical field drop between these two layers. Also, based on Nylon's chemical structure, the polarization should result in one side positive and the other side negative. Thus it is suggested the author double-check their working mechanism.
7. Overall, the capacitor charging curve is impressive. However, since the author mentioned that TENG is the 'most effective' energy harvester, the author should include the calculation of input mechanical energy, calculate an energy conversion ratio, and compare with other energy harvesting technology.
8. What is the limitation of such a TENG structure design?

Reviewer #2:

Remarks to the Author:

In the manuscript, the authors suggested a new strategy by designing shielding layer and alternative blank-tribo-area enabled charge space-accumulation (CSA) for boosting output performance of sliding mode triboelectric nanogenerator (CSA-S-TENG). As a result, the authors successfully demonstrated that the charge density of CSA-S-TENG achieves 1.63 mC/m^2 by designing grounded electrode to overcome the limitation of air breakdown and introducing extra blank-tribo area to accumulate charges. They validated working principles of device and demonstrated not only optimized CSA-S-TENG, but also their rotation type which drives hydro-thermometer by powering capacitor. There are several deficiencies to be revised, while I would recommend this manuscript for the possible publication in Nature Commun. after minor revision. The drawbacks and deficiencies are listed below.

- According to Equation (2) and Supplementary Note 1, transferred charge Q_t is affected by d_1 , d_2 , the thickness of dielectric film. However, the authors described and demonstrated that different thickness of dielectric film has no significant effect on the charges and output in the main text and

Supplementary Figure 11. Therefore, it is thought that equations do not explain phenomenon suitably or there are deficiencies in experimental validation. I recommend the authors to explain the reason why theoretical expectation and experimental result show different behavior.

- The authors described that dissipation of charges on the extra blank-tribo area enables maximizing charge space-accumulation effect. In Supplementary Figure 3, they demonstrated dissipation of charges occurs in tens of minutes and dissipation rate is not fast. However, in Supplementary Figure 4, the authors illustrated all the charges on the extra blank-tribo area are dissipated during even half cycle. Therefore, Supplementary Figure 4 is not exact to explain the role of extra blank-tribo area. It is thought that there is partial dissipation of charges on the extra blank-tribo area and it makes saturation of charges on the slider. Therefore, I recommend the authors to illustrate additionally the process of charge saturation of slider and partial dissipation of charges of extra blank-tribo area.

- Reference selection is basically okay. Addition of very recent comprehensive papers to reference section would make this manuscript better, e.g., Hinchet et al., *Science*, 365, 491, 2019.

Reviewer #3:

Remarks to the Author:

This manuscript reports a new strategy to boost the output performance of sliding mode TENG, by utilizing a shielding layer and alternative blank-tribo-area to enable charge space-accumulation. It is concluded that the shielding layer can effectively prevent the air breakdown occurring on interfacial layers, while the blank-tribo-area with rapid charge dissipation can promote charge accumulation. Authors performed a comprehensive study on the charge space-accumulation mechanism theoretically and experimentally, showing a 2.3 folds enhancement of normal S-TENG can be achieved. This study can be of interest to the broad audience in the community, thus I recommend the publication of this manuscript in *Nat Comm*, and have the following comments for the authors to address.

1. According to the author's description, the blank-area is designed for achieving charge space accumulation, and enabling the charge dissipation to replenish the charges. As per reviewer's understanding, the sliding speed is then affecting the amount of dissipated charges, and probably the charge space accumulation. Can the author comment on it?
2. In the working mechanism part, the authors stated that "the charges on the blank-tribo-area would be quickly dissipated, which ensures continuing charge replenishment on PTFE during each friction". In Figure S3, the graph shows that the charge on the PA layers decreases slowly, and a 50% drop may take a few minutes. This duration is much larger than the time interval between each sliding cycle. Can the author explain this phenomenon?
3. A few other strategies in previous literature to boost the output performance of TENG is suggested to be included in order to broaden the coverage of the introduction, e.g., *Nano Energy*, 2020, 73, 104760; *Adv. Sci.*, 6(24), 1901437; etc.
4. For the switch OFF state, what is the position of the slider when the switch is turned off? And what is the influence of switch-off position on the output performance?
5. What is the difference in designing a rotary mode CSA-S-TENG and a sliding mode CSA-S-TENG?
6. From Figure 2, the output charge increases from 200 nC to 700 nC with the increase of gap from 1.5 to 18.5 mm; while the charge increases from 400 nC to 700 nC when the extra tribo-area is extended from 46.5 to 76.5 mm. Apparently, the 17 mm increment of the gap contributes to the extra 500 nC charge, but the 30 mm increment on the tribo-area contributes to a 300 nC extra charge. Both gap and extra sliding range should correspond to the blank tribo-area. Could the author explain the different improvement efficiency of these two methods? Or is there an optimized ratio for maximum performance?
7. In Figure 2, the color of the dashed lines separating Figure 2 a-d is too light, especially closed to

white at the edges, which is hard to differentiate the four graphs. Also please label the outputs of different colors shown in Figure 2g and Figure 2h. For Figure 4b, the title of the x axis 'Time(s)' has been blocked, please check.

8. Although the charge boosting mechanism is designed for sliding mode TENG, can it also be used for other TENG modes to boost the output performance?

9. Environmental factors, such as humidity, have a great influence on the triboelectric output. Please discuss the influence of the humidity on the device's output.

Point-to-Point Response to the Reviewer's Comments

(Comments in black, response in blue)

Dear reviewers:

Thank you for your detailed and useful comments and valuable suggestions on our manuscript. We have revised the manuscript accordingly and the detailed corrections are listed below point by point.

Reviewer #1 (Remarks to the Author):

We thank the reviewer's detailed and responsible reviewing of our work. The triboelectric nanogenerator (TENG), which is based on the triboelectric effect and electrostatic induction, was invented in 2012 by Wang's group. Over the past seven years, in addition to versatile operation modes, the TENG has many other merits, including broad material availability, low cost, light weight and high efficiency even at low operation frequency (Adv. Energy Mater. 2018, 1802906). However, as a power source, low output charge density is one of the main barriers that prevent TENG from extensive application. In our work, we proposed a strategy of using the shielding electrode to avoid air breakdown in inner voids between two tribo-layers, and realizing the charge space-accumulation effect by further adding the alternative blank-tribo-area, which largely boosted the surface charge density and output power density of sliding mode TENG.

1. What does 'in-plane low frequency mechanical energy harvesting' mean? Why is it the 'most effective' method? What is the comparing metrics here? Please provide more detailed discussions.

Answer: Thanks for the reviewer's detailed reviewing. In ambient environment, there is much low frequency mechanical energy that is difficult to be harvested by

traditional electromagnetic power generation techniques, and these include **some sliding kinetic energy in the plane at low frequencies (<3Hz)**, including **reciprocating motion or rotation**, such as moving a mouse on the mouse pad (response to “what does ‘in-plane low frequency mechanical energy harvesting’ mean”). TENG have four different operation modes since its invention in 2012, including vertical contact-separation mode, single-electrode mode, lateral sliding mode and freestanding triboelectric-layer mode. Generally, TENG also can be divided into **vertical contact-separation mode and horizontal sliding mode** based on the driving modes. The vertical contact-separation mode TENG is better suited to harvesting vertical reciprocating motion, such as typing on a keyboard, walking, etc. **Compared with vertical contact-separation mode TENG**, sliding mode TENG could be **more efficient** for harvesting in-plane low frequency mechanical energy (response to “why is it the ‘most effective’ method? What is the comparing metrics here?”). For better understanding and avoiding misunderstandings, we have removed the word “most” in abstract).

Sliding mode triboelectric nanogenerator (S-TENG) is an effective technology for in-plane low-frequency mechanical energy harvesting.

2. The authors emphasized several times that TENG is the ‘most effective’, ‘most promising’ energy harvester in different applications. However, there is no data or chart to support this argument directly. Please benchmark using relevant to help readers understand it.

Answer: Thanks for this detailed reviewing. We do use “most effective” and “most promising” description once in the introduction. Existing mainstream technologies for harvesting mechanical energy include **electromagnetic generator (EMG), piezoelectric nanogenerator (PENG) and triboelectric nanogenerator (TENG)**. In previous literatures, the output performance of EMG and TENG at different

frequencies **has been systematically compared** (Adv. Mater., 2014, 26, 3580), (ACS Nano, 2016, 10, 4797). In general, the EMG has low voltage but a high output current while the TENG produces low output current but high output voltage. Due to the characteristics of mechanical structure, **EMG is suitable for harvesting high frequency and larger scale mechanical energy**, and low frequency energy is difficult for it to generate useful power. By using a three-dimensional intercalation electrode, Qin's group has improved the output of PENG to a new stage. However, **for each one unit, the output power of PENG is still smaller than TENG**, (Nat. Commun. 2020, 11, 1030). Relevant data were extracted in *Table R1* and *Table R2* for comparison (response to “there is no data or chart to support this argument directly”). At **low frequencies (<3Hz)**, lateral sling mode TENG is well suited to harvest the kinetic energy generated by these reciprocating motions due to its structure and adaptability. Therefore, for in-plane low frequency mechanical energy harvesting, the sliding mode TENG is the “most effective”, “most promising” energy harvester **compared with EMG and PENG** (response to “the authors emphasized several times that TENG is the ‘most effective’, ‘most promising’ energy harvester in different applications.” For better understanding and avoiding misunderstandings, we have removed the word “most” in introduction.).

Table R1 | The output of voltage and power density of EMG and TENG at different

Frequency	1 Hz		2 Hz		3 Hz	
Voltage/ Power density	Voltage (V)	Power density ($\mu\text{W}/\text{cm}^2$)	Voltage (V)	Power density ($\mu\text{W}/\text{cm}^2$)	Voltage (V)	Power density ($\mu\text{W}/\text{cm}^2$)
EMG	1	1.5E-02	1.5	1.5E-01	2.5	1.05
TENG	470	1.2	470	10	470	11

frequency (ACS Nano, 2016,10,4797).

Table R2 | Characteristics comparison of EMG and TENG under same conditions

(Adv. Mater., 2014, 26, 3580).

EMG		TENG	
Open-circuit voltage	67.6 mV	Short-circuit current	6.96 μ A
Internal resistance	12.3 Ω	Internal resistance	13.8 M Ω
Maximum power	102.6 μ W	Maximum power	140.4 μ W
Volume	288 cm ³	Volume	45.2 cm ³
Mass	412 g	Mass	53.3 g
Maximum power per unit volume	0.36 W/m ³	Maximum power per unit volume	3.11 W/m ³
Maximum power per unit mass	0.25 mW/kg	Maximum power per unit mass	2.63 mW/kg

Harvesting energy from ambient environment for self-powering distributed sensor networks has become a significant development direction in the Internet of Things (IoTs). Recently, based on the coupling of triboelectrification and electrostatic induction, triboelectric nanogenerator (TENG) with advantages of light weight, material variety, easy fabrication and low cost attracts great attention and has proved an efficient technology for harvesting low frequency mechanical energy such as human motion, wind, water wave and etc. Generally, TENG can be divided into vertical contact-separation mode (CS-) and horizontal sliding mode (S-) based on the driving modes. Different from CS-TENG, S-TENG holds high efficiency, continuous and high output for in-plane regular movement (e.g. reciprocation and rotation) conversion, and it is a promising one towards commercialization. Nevertheless, the low surface charge density is the bottleneck in the TENG output performance and its applications.

3. How good is ‘2.3 times’? How to evaluate this value? It is easy to boost the charge density from a tiny number to a relatively large number; however, it is challenging to improve the charge density starting from a big number.

Answer: We highly appreciate the reviewer for raising up this question. The output performance of TENG is **quadratic** to the charge density (Nat. Commun. 2015, 6, 8376), so it is important to improve charge density for boarding the applications of TENG. The charge density is about (50-430 μ C/m²) from 2012 to 2016 by material

selection and surface modification of tribo-layers and so on; The relevant researches in recent years achieve, $1003 \mu\text{C}/\text{m}^2$ in vacuum (Nat. Commun. 2017, 8, 88), $490\mu\text{C}/\text{m}^2$ in air by charge pumping, (Nat. Commun. 2018, 9, 3773), $1020\mu\text{C}/\text{m}^2$ in air by charge pumping, Nano Energy 2018, 49, 625), $1.25\text{mC}/\text{m}^2$ in air by charge excitation, (Nat. Commun. 2019, 10, 1426), $2.38 \text{ mC}/\text{m}^2$ in air by charge excitation and quantifying contact, (Nat Commun. 2020, 11, 1599,). Thus, the improvement of charge density is **very difficult** and **full of challenges** and a several hundred $\mu\text{C}/\text{m}^2$ increase is already very good. Currently, efforts to improve the charge density have been **concentrated in the contact-separated mode TENG**, and there has been **no substantial improvement** in the charge density output of the **sliding mode TENG** due to the presence of interface friction in the sliding TENG. Therefore, the 2.3-fold increase in charge density here is a **significant breakthrough** for improving the output performance of sliding mode TENG. (response to “how good is ‘2.3 times’? How to evaluate this value?”). Recently, the charge density of vertical contact-separation mode TENG has reached a very high level about $2.38 \text{ mC}/\text{m}^2$ (Nat. Commun. 2020, 11, 1599), and that of lateral sliding mode TENG also has reached a new level about $460 \mu\text{C}/\text{m}^2$ (Sci. Adv. 2019, 5, eaav6437). In this work, by optimizing tribo-materials and contact for normal S-TENG, the charge density with optimized parameters can reach $710 \mu\text{C}/\text{m}^2$ (which is much higher than previous report) and the ‘2.3 times’ enhancement obtained by our new design CSA-S-TENG is **compared with this value ($710 \mu\text{C}/\text{m}^2$)**, which is 3.54 times larger than $460 \mu\text{C}/\text{m}^2$ (previous record, Sci. Adv. 2019, 5, eaav6437). We highly agree with the reviewer that it would be easier to start with a lower value, but this work start with a high value, therefore, it is a great improvement. (response to “it is easy to boost the charge density from a tiny number to a relatively large number; however, it is challenging to improve the charge density starting from a big number.”).

4. Is this $1.63 \text{ mC}/\text{m}^2$ physically correct? Please justify this number by adding some physical explanation. What is the mechanism of keeping these charges on the surface,

and most importantly, how to effectively measure the value?

Answer: Thank the reviewer for these good questions. The effective charge density of **polytetrafluoroethylene** (PTFE) on the slider of CSA-S-TENG is 1.63 mC/m² and this value is true and reliable (response to “is this 1.63 mC/m² physically correct”), and the calculated method is **adopted from the previous work** (Adv. Mater. 2014, 26, 3788, Sci. Adv. 2019, 5, eaav6437). Due to air breakdown and imperfect contact electrification properties of material, there is **always existing saturated surface charge density** for common S-TENG (set as Q), which is usually small. Based on the space charge accumulation effect (analysis in the *Supplementary Note 3*), the amount of charge transferred through the external circuit can reach 2Q. These 2Q charges **can be used to power some electronics**, which is effective charges. As the slider moves on the stator (*Supplementary Figure 5a-c*), the charges distribution at the bottom electrodes change due to electrostatic induction, and 2Q charge will transfer from the right bottom electrode to the left bottom electrode. Therefore, the amount of effective transferred charge **depends on** the amount of charge on the tribo-layer of the slider (response to “adding some physical explanation”). Hence the effective charge density of PTFE (σ_{PTFE}) can be described as

$$\sigma_{PTFE} = \frac{Q_t}{S}$$

Where Q_t is the transferred charge between two bottom electrodes and S is the surface area of tribo-layer (PTFE).

Therefore, the effective charge density of the tribo-layer can be obtained **by measuring the amount of transferred charge between the bottom electrode or between the top electrode and the ground using the electrometer (Keithley 6514)** (response to “how to effectively measure the value”). In our designed CSA-S-TENG, the upper electrode on slider connects to the ground and there will be enough charge transferred from ground when the upper electrode needs. The surface charge of PTFE keeps **electrostatic equilibrium** with the lower tribo-layer, the bottom electrode and the upper shielding electrode, as shown in the *Supplementary Figure 5 d, e*. In

addition, based on a grounded conductive layer covered on the back of the slider, **air breakdown can be restrained to a great extent**. As a result, these charges can be kept on its surface (response to “what is the mechanism of keeping these charges on the surface”).

We have clarified the explanation in the text.

5. What is the scientific meaning of the FEA result in Fig. 1c? What is the boundary condition in the FEA initial setup? What is the contribution of the FEA result in this paper?

Answer: Thanks for the reviewer for raising up this question. The simulated results indicate **three times** potential difference between top (PTFE) layer and bottom (Nylon) layer **with and without** shielding layer, which could **greatly avoid air breakdown and hold larger charge density** on the tribo-layers (response to “what is the scientific meaning of the FEA result in Fig. 1c”). The thickness of the FEP and Nylon film are both 0.1 mm as well as the interval gap distance. the surface charge density of nylon and FEP set as $1\text{E-}6\text{ C/m}^2$ and $-1\text{E-}6\text{ C/m}^2$ respectively (response to “what is the boundary condition in the FEA initial setup”). The finite element analysis shows that the potential distribution in the air voids is significantly changed when the grounded electrode on sliding layer is introduced. And we **overcome the limitation of air breakdown** by designing grounded electrode in experiment according to this analysis (response to “what is the contribution of the FEA result in this paper”).

6. In Fig. 1b and also supporting Fig 1, the charge distribution in Nylon and metal is the same (both positive). Is it reasonable? If so, it means there is no electrical field drop between these two layers. Also, based on Nylon’s chemical structure, the polarization should result in one side positive and the other side negative. Thus it is suggested the author double-check their working mechanism.

Answer: We appreciate the reviewer for this detailed suggestion. As shown in *Figure 1b* and *Supplementary Figure 1a*, a traditional in-plane sliding mode TENG is composed of upper tribo-layer (PTFE), lower tribo-layer (Nylon) and bottom electrodes. When these two materials with opposite triboelectric polarities contact each other, due to the triboelectrification effect (or surface potential difference), **surface electrons transfer occurs** (Mater. Today, 2017, 20, 74). **The electrons will transfer from the surface of Nylon to PTFE**; thus, the positive charge distributes in the surface of Nylon and negative charge distribute in PTFE. In other words, the positive and negative charges appear on the surface of different materials rather than on the both sides of one material due to the charge transfer on contact interface. (response to “the polarization should result in one side positive and the other side negative”). The mount of positive charge in full surface of Nylon and negative charge in PTFE is **equal** due to the conservation of charge, and **the positive charge will distribute on the surface of Nylon uniformly after repeated sliding**. Whether the amount of charge on nylon and metal is the same or not, depends on **the number of bottom electrodes**. If there are only a pair of bottom electrodes (same as *Fig. 1b*), then it will be the same (response to “is it reasonable”). If there are more than one pair then it will be different, as shown in the **Figure R1b**.

Figure R1 | The charge distribution of traditional sliding mode TENG with one pair bottom electrode (a) and two pair bottom electrodes (b).

7. Overall, the capacitor charging curve is impressive. However, since the author mentioned that TENG is the ‘most effective’ energy harvester, the author should include the calculation of input mechanical energy, calculate an energy conversion ratio, and compared with other energy harvesting technology.

Answer: Thank the reviewer for raising up this question and the positive comment as “impressive”, and we are sorry that the word “most effective” confuses the reviewer. Previous work reported that freestanding TENG has a 100% theoretical conversion efficiency (Adv. Mater. 2014, 26, 2818) and the actual conversion efficiency is about 85% (Adv. Mater. 2014, 26, 6599), and sliding mode TENG has 50% actual conversion efficiency (Adv. Mater. 2014, 26, 3788). We calculated the conversion efficiency of CSA-S-TENG in this work to be about **36~48%** according to **the calculation method above**, and the energy consumption here should be considered as the sum of all parts. The main energy consume is from friction and mechanical motion, therefore, the energy conversion efficiency will increase appropriately along with the increase of charge density. **Previous work has systematically compared TENG and EMG, demonstrating that TENG has a higher conversion efficiency than EMG at low frequencies** (ACS Nano, 2016, 10, 4797, Adv. Mater., 2014, 26, 3580, Nanotechnology, 2014, 25, 135402). We are sorry that this work does not hybridize with EMG or combine with PENG, so direct comparison cannot be made.

8. What is the limitation of such a TENG structure design?

Answer: Thanks for the reviewer for raising up this question. This charge space accumulation structure is only suitable to sliding mode TENG now, which can be used to harvest sliding or rotation energy just. The charge dissipated tribo-materials can further enhance the output, however, it also **limits the material selection rang** for CSA-S-TENG.

Reviewer #2 (Remarks to the Author):

In the manuscript, the authors suggested a new strategy by designing shielding layer and alternative blank-tribo-area enabled charge space-accumulation (CSA) for boosting output performance of sliding mode triboelectric nanogenerator (CSA-S-TENG). As a result, the authors successfully demonstrated that the charge density of CSA-S-TENG achieves 1.63 mC/m² by designing grounded electrode to overcome the limitation of air breakdown and introducing extra blank-tribo area to accumulate charges. They validated working principles of device and demonstrated not only optimized CSA-S-TENG, but also their rotation type which drives hydro-thermometer by powering capacitor. There are several deficiencies to be revised, while I would recommend this manuscript for the possible publication in Nature Commun. after minor revision. The drawbacks and deficiencies are listed below.

Answer: We highly appreciate the reviewer's positive comments on our work. And we also thank the reviewer's detailed and responsible reviewing of our work.

- According to Equation (2) and Supplementary Note 1, transferred charge Q_t is affected by d_1 , d_2 , the thickness of dielectric film. However, the authors described and demonstrated that different thickness of dielectric film has no significant effect on the charges and output in the main text and Supplementary Figure 11. Therefore, it is thought that equations do not explain phenomenon suitably or there are deficiencies in experimental validation. I recommend the authors to explain the reason why theoretical expectation and experimental result show different behavior.

Answer: Thank reviewer for the detailed reviewing. For the traditional sliding mode TENG (S-TENG), when the upper electrode grounded, we can get the **saturated**

charge of top tribo-layer Q_{0g} as follow according to our theoretical analysis

$$Q_{0g} = \frac{2Q \left(\frac{d_2 \varepsilon_{r1} + 1}{d_1 \varepsilon_{r2}} \right)}{\frac{d_2 \varepsilon_{r1} + 2}{d_1 \varepsilon_{r2}}} = 2Q \left(1 - \frac{1}{\frac{d_2 \varepsilon_{r1} + 2}{d_1 \varepsilon_{r2}}} \right)$$

Where ε_{r1} , d_1 and ε_{r2} , d_2 are the relative permittivity and the thickness of the top and bottom dielectric layer respectively, and Q is saturated effective surface charge on the top tribo-layer of ungrounded traditional S-TENG.

As traditional S-TENG evolved into S-TENG with charge space accumulation structure, based on the analysis of charge transfer, the **transferred charge** between the bottom electrodes (Q_t) is determined by the following formula

$$Q_t = \frac{3}{4} Q_{0g} + Q = \frac{Q}{2} \left(5 - \frac{3}{\frac{d_2 \varepsilon_{r1} + 2}{d_1 \varepsilon_{r2}}} \right)$$

From the above formulas, it can be concluded that when the thickness of the upper tribo-layer increases, the effective surface charge of the upper tribo-layer decreases, and the transferred charge between the bottom electrodes should also decreases for S-TENG (**without considering charges dissipation**). However, in actual experiments, the charges in the extra blank-tribo-area would inevitably dissipate into air (charge dissipation) due to the strong electric filed on its surface. Therefore, there is a dynamic equilibrium process that between dissipation and replenishment and charge dissipation **further enhances the charge density**, which leads the distinction between experimental and theoretical results. During this process, it is difficult to be fully described by simple formula owing the lack of method to evaluate the dissipating ability of materials. When the thickness of the upper tribo-layer increases, the effective charge on its surface is relatively low. But due to the ability of the extra blank-tribo-area to **replenish the charge**, partial surface charge can be **recovered** to a certain extent, so that the transferred charge between the bottom electrodes is less affected by the film thickness (as shown in *Supplementary Figure 11*). And we have revised the manuscript.

Furthermore, the different thickness of PTFE within a certain range has little effect on the output due to the replenishment ability of extra blank-tribo-area (as shown in **Supplementary Figure 11**). Therefore, thicker dielectric materials can be used as the triboelectric layer for durability. Thus, for the slider material, we would better choose charge-keeping property and higher surface charge state density, and for the stator material, we need a faster charge dissipating feature to maximize charge space-accumulation effect. The output of both single and double bottom electrode CSA-S-TENG can be optimized (**Supplementary Figure 12, 13**).

- The authors described that dissipation of charges on the extra blank-tribo area enables maximizing charge space-accumulation effect. In Supplementary Figure 3, they demonstrated dissipation of charges occurs in tens of minutes and dissipation rate is not fast. However, in Supplementary Figure 4, the authors illustrated all the charges on the extra blank-tribo area are dissipated during even half cycle. Therefore, Supplementary Figure 4 is not exact to explain the role of extra blank-tribo area. It is thought that there is partial dissipation of charges on the extra blank-tribo area and it makes saturation of charges on the slider. Therefore, I recommend the authors to illustrate additionally the process of charge saturation of slider and partial dissipation of charges of extra blank-tribo area.

Answer: Thanks for reviewer's good question and valuable suggestion. **Sufficient triboelectrification** was applied to the surface of the Nylon film which on the Al-PA single-electrode CS-TENG and Al-PA-Al double-electrode CS-TENG before measurement by using Aluminum to contact with Nylon for triboelectrification 30 s. Therefore, **the initial surface charge density of Nylon should be the same for the both modes CS-TENG**. There is a period of time (about 1-2 s) between the experimental conditions being ready and the beginning of the measurement due to the practical operation of the experiment, and **during this time** the charge dissipates into

the air too. Therefore, it is hard to know its initial charge surface density, but according to *Supplementary Figure 3b*, the value is **at least 50 $\mu\text{C}/\text{m}^2$** . But in the first time we measure Al-PA single-electrode CS-TENG, the surface charge density is 17 $\mu\text{C}/\text{m}^2$, suggesting that the surface charge had dissipated by **at least 66%** (33 $\mu\text{C}/\text{m}^2$) and it is really fast. In addition, the charge density dropped 25% for the second cycle of Al-PA single-electrode CS-TENG, but only 13% for Al-PA-Al double-electrode CS-TENG (response to “dissipation of charges occurs in tens of minutes and dissipation rate is not fast”). The charge accumulation process of CSA-S-TENG will go on for several cycles (about 30 cycles) as shown in *Figure 1j* (the third case). As the charges on the extra blank-tribo-area dissipate, the surface charge density of the upper tribo-layer gradually reaches saturation and the transferred charge between the bottom electrode increase and reach the maximum output. To express this process **concisely**, we present the principle in one cycle in *Supplementary Figure 4*. However, it **does not mean** that all the charges in the extra blank-tribo-area will dissipate in half a cycle. We are sorry for the confusion caused to the reviewer. In addition, **not all surface charges** on the extra blank-tribo-area dissipate into the air **at once**, and there are **still few charges on it** (as shown in *Supplementary Figure 2b*) due to the balance between charge induction and dissipation and this is a dynamic equilibrium process that is difficult to visualize in a schematic (response to “illustrate additionally the process of charge saturation of slider and partial dissipation of charges of extra blank-tribo area.”). We have revised the manuscript and the supplementary material.

It is worth noting that, without the bottom electrode for equilibrating electrostatic field and the feature of PA material, the most charges on the blank-tribo-area would be quickly dissipated and few charges left, which ensures continuing charge replenishment on PTFE during each friction (**Supplementary Figure 2, 3**), so the CSA-S-TENG can easily achieve stable and multifold output charge. The process of this charge space-accumulation is analyzed in detail and presented in **Supplementary Figure 4** and **Supplementary Note 2**, and the charge accumulation process of

CSA-S-TENG will go on for several cycles.

Supplementary Figure 3a, c show the charge density for a single electrode mode contact-separated TENG with 50 μm PA and PTFE thin films. Sufficient triboelectrification was applied to the surface of the PA and PTFE film before measurement. Therefore, the initial surface charge density should be the same for the both modes CS-TENG. There is a period of time (about 1-2 s) between the experimental conditions being ready and the beginning of the measurement due to the practical operation of the experiment, and during this time the charge dissipates into the air too. Therefore, it is hard to know its initial charge surface density of PA, but according to Supplementary Figure 3b, the value is at least $50 \mu\text{C}/\text{m}^2$. But in the first time we measure Al-PA single-electrode CS-TENG, the surface charge density is $17 \mu\text{C}/\text{m}^2$, suggesting that the surface charge had dissipated by at least 66% ($33 \mu\text{C}/\text{m}^2$). The charge density of double electrode mode CS-TENG are four times higher than that of single electrode mode. This demonstrates that charges are more easily dissipated into air when there are no electrodes under triboelectric layer, while the charges can keep on that with electrodes underneath due to the shielding function of electrode. In other words, the charges on triboelectric layer are bounded by the charges on electrode.

- Reference selection is basically okay. Addition of very recent comprehensive papers to reference section would make this manuscript better, e.g., Hinchet et al., Science, 365, 491, 2019.

Answer: Thanks for the reviewer's good suggestion. The reference is great and we have revised the manuscript accordingly and cited this paper.

17. Hinchet R., et al., Transcutaneous ultrasound energy harvesting using capacitive triboelectric technology. Science 365, 491-494 (2019).

Reviewer #3 (Remarks to the Author):

This manuscript reports a new strategy to boost the output performance of sliding mode TENG, by utilizing a shielding layer and alternative blank-tribo-area to enable charge space-accumulation. It is concluded that the shielding layer can effectively prevent the air breakdown occurring on interfacial layers, while the blank-tribo-area with rapid charge dissipation can promote charge accumulation. Authors performed a comprehensive study on the charge space-accumulation mechanism theoretically and experimentally, showing a 2.3 folds enhancement of normal S-TENG can be achieved. This study can be of interest to the broad audience in the community, thus I recommend the publication of this manuscript in Nat Comm, and have the following comments for the authors to address.

Answer: We highly appreciate the reviewer's positive comments on our work as "be of interest to the broad audience". And we also thank the reviewer's detailed and responsible reviewing of our work.

1. According to the author's description, the blank-area is designed for achieving charge space accumulation, and enabling the charge dissipation to replenish the charges. As per reviewer's understanding, the sliding speed is then affecting the amount of dissipated charges, and probably the charge space accumulation. Can the author comment on it?

Answer: Thank reviewer for the detailed reviewing. The sliding speed affects **the time** of the extra blank-tribo-area **exposed to air** in one cycle and **the amount of dissipated charges** due to the fact that not all surface charges on the extra blank-tribo-area dissipate into the air at once. The exposed time of the extra blank-tribo-area to air in one cycle decreases as the sliding speed increases. As the

charge accumulation process of CSA-S-TENG goes on **for several cycles** (as shown in *Figure 1j*, the third case), the process of charge accumulation **may take more cycles in a higher speed**, but the output still can reach same saturation, which proves the same charge quantity in different speeds in *Figure 2d*.

2. In the working mechanism part, the authors stated that “the charges on the blank-tribo-area would be quickly dissipated, which ensures continuing charge replenishment on PTFE during each friction”. In *Figure S3*, the graph shows that the charge on the PA layers decreases slowly, and a 50% drop may take a few minutes. This duration is much larger than the time interval between each sliding cycle. Can the author explain this phenomenon?

Answer: Thank reviewer for the detailed reviewing. In *Supplementary Figure 3*, sufficient triboelectrification was applied to the surface of the Nylon film which on the Al-PA single-electrode CS-TENG and Al-PA-Al double-electrode CS-TENG before measurement by using Aluminum to rub with Nylon for 30 s. Therefore, the initial surface charge density on Nylon should be **the same** for the both modes CS-TENG. There is **a period of time** (about 1-2 s) between the experimental conditions being ready and the beginning of the measurement due to the practical operation of the experiment, and during the time the **charge dissipates into the air too**. Therefore, it is hard to know its initial charge surface density, but according to *Supplementary Figure 3b*, the value is at least $50 \mu\text{C}/\text{m}^2$. But in the first time we measure Al-PA single-electrode CS-TENG, the surface charge density is $17 \mu\text{C}/\text{m}^2$, **suggesting that the surface charge had dissipated by at least 66% ($33 \mu\text{C}/\text{m}^2$ within 2 seconds) and it is really fast**. In addition, the charge density dropped 25% for the second measurement of Al-PA single-electrode CS-TENG, but **only 13% for Al-PA-Al double-electrode CS-TENG**. Therefore, the charge accumulation process of CSA-S-TENG will go on for several cycles. We have revised the manuscript and the supplementary material.

It is worth noting that, without the bottom electrode for equilibrating electrostatic field and the feature of PA material, the most charge on the blank-tribo-area would be quickly dissipated, which ensures continuing charge replenishment on PTFE during each friction (**Supplementary Figure 2, 3**), so the CSA-S-TENG can easily achieve stable and multifold output charge. The process of this charge space-accumulation is analyzed in detail and presented in **Supplementary Figure 4** and **Supplementary Note 2**, and the charge accumulation process of CSA-S-TENG will go on for several cycles.

Supplementary Figure 3a, c show the charge density for a single electrode mode contact-separated TENG with 50 μm PA and PTFE thin films. Sufficient triboelectrification was applied to the surface of the PA and PTFE film before measurement. Therefore, the initial surface charge density should be the same for the both modes CS-TENG. There is a period of time (about 1-2 s) between the experimental conditions being ready and the beginning of the measurement due to the practical operation of the experiment, and during which time the charge dissipates into the air too. Therefore, it is hard to know its initial charge surface density of PA, but according to Supplementary Figure 3b, the value is at least $50 \mu\text{C}/\text{m}^2$. But in the first time we measure Al-PA single-electrode CS-TENG, the surface charge density is $17 \mu\text{C}/\text{m}^2$, suggesting that the surface charge had dissipated by at least 66% ($33 \mu\text{C}/\text{m}^2$). The charge density of double electrode mode CS-TENG are four times higher than that of single electrode mode. This demonstrates that charges are more easily dissipated into air when there are no electrodes under triboelectric layer, while the charges can keep on that with electrodes underneath due to the shielding function of electrode. In other words, the charges on triboelectric layer are bounded by the charges on electrode.

3. A few other strategies in previous literature to boost the output performance of

TENG is suggested to be included in order to broaden the coverage of the introduction, e.g., *Nano Energy*, 2020, 73, 104760; *Adv. Sci.*, 6(24), 1901437; etc.

Answer: Thanks for the reviewer's good suggestion, and we have revised the manuscript.

Recently, Zhu et al. reported a direct current TENG by using charges unidirectional transportation and dual-intersection TENG, and successfully realized a continuous motion control in virtual space for next-generation real-time VR application in triboelectric⁴⁰. Liu et al. reported a constant current S-TENG and indicated that air breakdown effect would happen on the sliding edge of two tribo-layers. By directly utilizing the discharged charges and improving contact, this kind of device reached $460 \mu\text{C}/\text{m}^2$ charge density compared with $70 \mu\text{C}/\text{m}^2$ from controlled S-TENG device⁴¹. In most cases of S-TENG, air breakdown happens not only on the edge of sliding layer, but also in the overlapped interface due to inescapable air voids between two osculatory tribo-layers⁴².

11. He T., et al., Self-Sustainable Wearable Textile Nano-Energy Nano-System (NENS) for Next-Generation Healthcare Applications. *Adv. Sci.* **6**, 1901437 (2019).

40. Zhu J., et al., Continuous direct current by charge transportation for next-generation IoT and real-time virtual reality applications. *Nano Energy* **73**, 104760 (2020).

4. For the switch OFF state, what is the position of the slider when the switch is turned off? And what is the influence of switch-off position on the output performance?

Answer: Thanks for the reviewer for raising up this question. When the switch is turn

OFF, the slider is **on the left extra blank-tribo-area** (response to “what is the position of the slider when the switch is turned off”). Because of the huge surface potential, the charge on the upper electrode and the tribo-layer cannot remain intact. And **no matter where the position of slider**, when the switch is turned off, the output will sharp decline because the charge space accumulation effect is broken (response to “what is the influence of switch-off position on the output performance”).

5. What is the difference in designing a rotary mode CSA-S-TENG and a sliding mode CSA-S-TENG?

Answer: Thanks for the reviewer for raising up this question. Rotary mode CSA-S-TENG are realized by **radially arraying** of basic units of sliding mode CSA-S-TENG, the **shape** of electrode and blank area also needs to be changed from rectangle to sector, and the structure of each unit is **similar** to that of a sliding mode CSA-S-TENG.

6. From Figure 2, the output charge increases from 200 nC to 700 nC with the increase of gap from 1.5 to 18.5 mm; while the charge increases from 400 nC to 700 nC when the extra tribo-area is extended from 46.5 to 76.5 mm. Apparently, the 17 mm increment of the gap contributes to the extra 500 nC charge, but the 30 mm increment on the tribo-area contributes to a 300 nC extra charge. Both gap and extra sliding range should correspond to the blank tribo-area. Could the author explain the different improvement efficiency of these two methods? Or is there an optimized ratio for maximum performance?

Answer: Thanks for your good question and valuable suggestion. The **gap** between the two bottom electrodes is **more affected** by the electric field of the bottom electrodes than the extra blank-tribo-area on either side. Therefore, even both gap and extra sliding range correspond to the blank tribo-area, but the optimized ratio for

maximum performance are different, **26.23 and 20.68 nC/mm** for gap and extra sliding range, respectively. And we have revised the manuscript.

From the test results in **Figure 2a**, with the increase of gap distance from 1.5 to 18.5 mm, the output charge increases linearly from 200 nC to 700 nC for S-TENG with the shielding electrode and quickly becomes saturated from 200 nC to 400 nC for S-TENG without the shielding electrode. Furthermore, as shown in **Figure 2b**, with fixed gap distance, the increase of blank-tribo-area and sliding range on the outside area of the two bottom electrodes also leads to the enhancement of charge space-accumulation effect in CSA-S-TENG. With the extra tribo-area extending from 46.5 to 76.5 mm, the output charge and current of CSA-S-TENG increase from 400 nC, 0.8 μ A to 700 nC, 1.6 μ A linearly. And the optimized ratio for maximum performance are different, 26.23 and 20.68 nC/mm for gap and extra sliding range, respectively, due to the different influence caused by the electric field of bottom electrodes.

7. In Figure 2, the color of the dashed lines separating Figure 2 a-d is too light, especially closed to white at the edges, which is hard to differentiate the four graphs. Also please label the outputs of different colors shown in Figure 2g and Figure 2h. For Figure 4b, the title of the x axis 'Time(s)' has been blocked, please check.

Answer: Thank reviewer for the detailed reviewing and good suggestion. We **deepened** the color of the dashed line in *Figure 2*, and **labeled** the outputs of different colors shown in *Figure 2g* and *Figure 2h*. And we have **adjusted** the position of *Figure 4b* and the 'Time(s)' shows clearly.

Figure 2 | Structure and materials influence on the output of CSA-S-TENG. a Output charge of double bottom electrode S-TENG with different inner electrode gaps when grounded switch OFF/ON. **b** Output charge and current of CSA-S-TENG with fixed electrode gap while varying sliding range (speed fixed at 4 cm/s). **c** Transferred charge of CSA-S-TENG with different inner electrode gaps when grounded switch OFF/ON. **d** Output charge and current of CSA-S-TENG with fixed bottom electrode gap while varying sliding speed (range fixed at 76.5 mm). **e,f** Output charge of single electrode CSA-S-TENG with fixed stationary material PA while varying sliding materials (FEP, PTFE and Kapton) and fixed sliding material PA while varying stationary materials (FEP, PTFE and Kapton) respectively. Optimized output charge density of **g** double electrode and **h** single electrode CSA-S-TENG with grounded switch OFF/ON.

Figure 4 | Performance and application of rotation-type CSA-S-TENG. **a** 3D structural schematic of the rotary device. **Inset 1** and **2** respectively depict the top view schematics and device photographs of stator and rotator part. Scale bar: 1 cm. **b** Short circuit charge, current and open-circuit voltage of rotary CSA-S-TENG at 0.5 Hz working speed. **c** Transferred charge and current of rotary CSA-S-TENG at different rotational speed. **d** Matching impedance and output power evaluation of rotary CSA-S-TENG at 1 Hz working speed. **e** Voltage curves of charging 100 μF , 470 μF and 1 mF capacitor using rotary CSA-S-TENG at 1 Hz speed. **f** Directly driving 912 LEDs at 60 rpm. **g** Charging 22 μF capacitor while powering a scientific calculator at 110 rpm. **h** Charging 470 μF capacitor while powering a hydro-thermometer at 120 rpm.

8. Although the charge boosting mechanism is designed for sliding mode TENG, can it also be used for other TENG modes to boost the output performance?

Answer: Thanks for the reviewer for raising up this question. Charge space accumulation structure can be well applied to sliding or rotary mode TENG, to improve its charge density, which might be further optimized and applied in other mode TENG.

9. Environmental factors, such as humidity, have a great influence on the triboelectric output. Please discuss the influence of the humidity on the device's output.

Answer: Thanks for the reviewer for this valuable suggestion. We measured all the output performance **in room conditions**. Different from the contact-separation mode TENG, due to the characteristics of the S-TENG, **the slider is always in close contact with the tribo-layer of stator**, so the humidity has little influence on the slider. Since the surface charge dissipation of the extra tribo-layer is **beneficial** to charge accumulation, humidity has little influence on CSA-S-TENG. We have added new experiment results in different humidity as shown in *Supplementary Figure 17* and revised the manuscript.

Furthermore, an excellent humidity adaptability is also verified for CSA-S-TENG by measuring output charge in different humidity from 10% to 80% RH (**Supplementary Figure 17**).

Supplementary Figure 17 | Transferred charge of CSA-S-TENG in different humidity. Due to the characteristics of the S-TENG, the slider is always in close contact with the tribo-layer of stator, so the humidity has little influence on the slider. Since the surface charge dissipation of the extra tribo-layer is beneficial to charge accumulation, humidity has little influence on CSA-S-TENG.

Reviewers' Comments:

Reviewer #1:

Remarks to the Author:

1, About Q6 and Fig. R1, the authors did not address the question raised. The authors only gave an explanation of the working mechanism of TENG, which is not related to Q6 in my previous comments. Indeed, the amount of charge in Nylon and PTFE should be the same if surface charge dissipation is safely ignored. However, why is the amount of charge, regardless of symbol, in Nylon and metal (left) the same? Based on Nano Lett. 2013, 13, 5, 2226–2233, when the polymer (Nylon and PTFE) has no complete overlap, there will be charge induced in the metal, which is due to the charge conservation in the whole system. However, what the authors showed in the manuscript, even when Nylon and PTFE have complete overlap, the metal has charges (leaving alone the number and symbol), which might violate the basic working mechanism of sliding-mode TENG. It is suggested the authors should justify this conclusion. Also, it is suggested the authors should calculate the electrical field drop in Nylon to prove that the Nylon layer did not fully screen the electrical field.

2, Following up the above comment. The authors mentioned that the reason Nylon has surface charge is due to the electron transfer, which is reasonable. However, if there is no polarization in Nylon, how to explain that the metal can also get charged? If it is based on electrostatic induction, where does the induction come from if Nylon has no polarization?

3, In Fig. R1-a and b, the authors equally 'divide' the charge based on the number of the metal. Does that mean the charge distribution in Nylon also perfectly aligns well with metal? If it is a theoretical statement, it is suggested the authors should provide more details on the relationship between the number of metals and its charge density.

Reviewer #2:

Remarks to the Author:

The authors took into consideration the reviewer's comments, they analyzed additional measurements and answered to the questions. They addressed most of reviewers' comments and took them into account by modifying the manuscript. They nicely answered the questions by providing data, analysis and discussions. The manuscript has been changed and is now better and clearer. Overall I think this work can now be published in Nature Comm. This is why I recommend accepting this manuscript.

Reviewer #3:

Remarks to the Author:

The authors have addressed all the concerns from the reviewers. I suggest the acceptance now.

Point-to-Point Response to the Reviewer's Comments
(Comments in black, response in blue)

Reviewer #1 (Remarks to the Author):

1, About Q6 and Fig. R1, the authors did not address the question raised. The authors only gave an explanation of the working mechanism of TENG, which is not related to Q6 in my previous comments.

Indeed, the amount of charge in Nylon and PTFE should be the same if surface charge dissipation is safely ignored. However, why is the amount of charge, regardless of symbol, in Nylon and metal (left) the same? Based on Nano Lett. 2013, 13, 5, 2226–2233, when the polymer (Nylon and PTFE) has no complete overlap, there will be charge induced in the metal, which is due to the charge conservation in the whole system. However, what the authors showed in the manuscript, even when Nylon and PTFE have complete overlap, the metal has charges (leaving alone the number and symbol), which might violate the basic working mechanism of sliding-mode TENG. It is suggested the authors should justify this conclusion. Also, it is suggested the authors should calculate the electrical field drop in Nylon to prove that the Nylon layer did not fully screen the electrical field.

Answer: We highly appreciate the reviewer for the detailed and responsible reviewing of our work. We are sorry that the previous answers didn't completely dispel reviewer's doubts and we will continue to elaborate more carefully. The structure of the traditional sliding mode TENG in our work is slightly different from the structure of the sliding mode TENG in the Nano Lett. 2013, 13, 5, 2226–2233. The slider and stator of the TENG in Nano Lett. 2013, 13, 5, 2226–2233 have a symmetrical structure, differing in the thin film on each surface. Therefore, the area of the film (PTFE and Nylon) on the surface of the stator is equal to that on the surface of the slider. Due to the conservation of charge, after triboelectrification, the amount of negative charge on the surface of PTFE is equal to the amount of positive charge on

the surface of Nylon, and the surface charge density is also equal. When Nylon and PTFE have complete overlap, the positive charge on the Nylon balances with the negative charge on the surface of PTFE, therefore, there are no more net charge and there will be no electric potential drop across the two electrodes, so there is no charge on the metal as shown in Figure R2a, i. When the slider and stator separate, there is charge on metals due to the electrostatic induction and the connection between top electrode and bottom electrode. However, the slider of the traditional sliding mode TENG in our work without the top electrode and its area is much smaller than stator. Therefore, the amount of negative charge on the surface of PTFE is still equal to that of Nylon surface, but the surface charge density is not equal (the surface charge density of PTFE is higher than that of Nylon). When the left metal in stator and the slider completely overlap, the positive charge on the left part of Nylon cannot completely balances with the all negative charge on the PTFE and needs more positive charge to balance, so there is positive charge on the left metal due to the electrostatic induction (response to “even when Nylon and PTFE have complete overlap, the metal has charges (leaving alone the number and symbol)”). We are sorry that we have to correct the error about the sign of charge on the Nylon in the Figure R1b and present it below. The area of the PTFE on the slider of the traditional sliding mode TENG in our work is half of the area of the Nylon on the stator. Due to the conservation of charge, after triboelectrification, the amount of negative charge on the surface of PTFE is equal to the amount of positive charge on the surface of Nylon, thus the surface charge density of Nylon is half of that of PTFE. When the left metal and the slider completely overlap, only half negative charge balances with the positive charge on the Nylon, and the rest of half negative charge induces half of the positive charge on the left metal as shown in Figure R1a, so the amount of charge in Nylon and metal both are half of the all positive charge and equal (response to “why is the amount of charge, regardless of symbol, in Nylon and metal (left) the same”). In the TENG model shown in Figure R1a, the amount of surface charge of PTFE is set as Q and the area of the PTFE and left metal is set as S . So

$$\sigma_0 = \frac{Q}{S} \quad (1)$$

Where the σ_0 is the surface charge density of PTFE.

Ignore the edge effect of electric field, according to Gauss's theorem

$$E_0 = \frac{\sigma_0}{2\varepsilon_0} = \frac{Q}{2S\varepsilon_0} \quad (2)$$

Where the E_0 is the electric field generated by the charge of PTFE, the ε_0 is vacuum dielectric constant.

Similar, the electric field generated by the charge of Nylon E_p

$$E_p = \frac{\sigma_p}{2\varepsilon_0} = \frac{Q}{4S\varepsilon_0} \quad (3)$$

Where the σ_p is the surface charge density of Nylon.

Therefore, the electric field in Nylon E_N

$$E_N = E_0 - E_p = \frac{Q}{4S\varepsilon_0} \quad (4)$$

To balance this electric field, the $1/2 Q$ charge will be induced in the left metal (response to “it is suggested the authors should calculate the electrical field drop in Nylon to prove that the Nylon layer did not fully screen the electrical field.”).

Figure R1 | The charge distribution of traditional sliding mode TENG with one pair bottom electrode (a) and two pair bottom electrodes (b).

Figure R2 | **a** The charge distribution of the sliding mode TENG with top electrode and one bottom electrode (Nano Lett. 2013, 13, 5, 2226–2233), **b** The charge distribution of traditional sliding mode TENG with one pair bottom electrode and none of top electrode.

2, Following up the above comment. The authors mentioned that the reason Nylon has surface charge is due to the electron transfer, which is reasonable. However, if there is no polarization in Nylon, how to explain that the metal can also get charged? If it is based on electrostatic induction, where does the induction come from if Nylon has no polarization?

Answer: Thank the reviewer for raising up this question. In Figure R1a, due to the surface area of nylon layer is two times larger than that of PTFE layer, therefore, the surface charge density or total charge quantity on PTFE surface is two times larger than nylon layer with the same surface area. In this case, there will be $1/2 Q$ on PTFE balanced with overlapped nylon, and $1/2 Q$ on left metal will be induced by the rest unbalanced charge on PTFE. Hence, there are charge on the left metal.

3, In Fig. R1-a and b, the authors equally ‘divide’ the charge based on the number of the metal. Does that mean the charge distribution in Nylon also perfectly aligns well with metal? If it is a theoretical statement, it is suggested the authors should provide more details on the relationship between the number of metals and its charge density.

Answer: Thank the reviewer for raising up this question and good suggestion. We are sorry about the error about the sign of charge on the Nylon in the Figure R1b and the mistake has been corrected. The charge distribution in Nylon depends on the triboelectrification between Nylon and PTFE and its surface charge density depends on the total surface charge and its area. In the Figure R1b, we fixed the area of the slider. In addition to adding the bottom electrode pair, the area of stator was extended to match the slider. The surface charge of slider still set as Q . Therefore, the total charge quantity on Nylon will not change, but the charge on the metal will change due to the diminution of the charge density of Nylon. In this situation, if there are N bottom electrode pairs, the charge on the metal overlapped by the slider will be $+(2N-1)/(2N) Q$, while the charge on the other metal will be $-1/(2N) Q$.

Reviewer #2 (Remarks to the Author):

The authors took into consideration the reviewer’s comments, they analyzed additional measurements and answered to the questions. They addressed most of reviewers’ comments and took them into account by modifying the manuscript. They nicely answered the questions by providing data, analysis and discussions. The manuscript has been changed and is now better and clearer. Overall I think this work can now be published in Nature Comm. This is why I recommend accepting this manuscript.

Response: Thanks for your positive comment.

Reviewer #3 (Remarks to the Author):

The authors have addressed all the concerns from the reviewers. I suggest the acceptance now.

Response: Thanks for your positive comment.